# MANGO: Mangrove Global Observations – A Dataset and Benchmark

## Abstract

Mangroves buffer coasts and store large amounts of carbon, yet they are vulnerable to storms and require reliable monitoring at global scale. Thresholded spectral indices break across sensors, seasons, and atmospheres, which limits their usefulness beyond local settings. Recent segmentation models are more promising but are difficult to train at scale because single-date imagery and labels are rarely paired and because models seldom exploit location context. First, we collect a globally distributed dataset, *MANGO*, that pairs one Sentinel-2 acquisition with each region–year label through a principled selection that balances agreement with the label and scene quality, and we provide country-disjoint splits together with co-registered geospatial embeddings. Second, we introduce a simple way to turn a global geospatial embedding into a small set of context channels that augment the optical bands and condition any backbone without architectural changes. Across strong convolutional and transformer baselines, this combination yields consistent gains on held-out countries and visibly cleaner maps, with sharper shorelines, better retention of small stands, and fewer false positives over turbid water, while adding minimal computational overhead. We release the dataset, the selection protocol, and the conditioning module to support reliable and scalable monitoring of coastal ecosystems.

## 1 Introduction

Mangroves are keystone coastal forests that buffer shorelines, sustain fisheries, and store substantial blue carbon, therefore monitoring at global scale is essential Bunting et al. (2022); Cissell et al. (2021). Operational pipelines span thresholded spectral indices such as the Normalized Difference Vegetation Index (NDVI) Tran et al. (2022), the Modified Normalized Difference Water Index (MNDWI) Tran et al. (2022), and the Mangrove Vegetation Index (mvi) Baloloy et al. (2020), as well as machine learning workflows on Google Earth Engine (GEE) Tran et al. (2022). These index-based approaches are not robust across sensors and seasons and are sensitive to water adjacency and atmospheric effects, which often forces site-specific retuning Tran et al. (2022); Baloloy et al. (2020). This fragility is evident in Figure 1: panels (a) and (d) are two Sentinel-2 images of the same region on different dates, panels (b) and (e) visualize mvi after clipping the upper one percent to improve contrast, and a single mvi threshold tuned in the range 4.5–13 extracts mangroves on the first date as in panel (c) yet fails on the second date as in panel (f) despite identical location.

To move beyond the fragility of index thresholds, recent studies pursue supervised deep learning for mangrove mapping and report steady gains across diverse coastlines Fu et al. (2025); Xu et al. (2023). Yet in practice many studies train with Global Mangrove Watch(GMW) data Bunting et al. (2022). GMW provides region–year masks, one annual mask per mapped region and year that summarizes many observations, yet it does not specify the acquisition date of a single optical scene Bunting et al. (2022). A single-date Sentinel-2 image therefore often fails to align with the annual label in time and tide, and naïve pairing introduces label noise. In response, many works fall back to small local datasets or couple annual masks with mismatched imagery or temporal composites, which increases manual quality control and undermines reproducibility and cross-region comparability Cissell et al. (2021).

We address this issue by providing *MANGO*, a collection of Sentinel-2 Level-2A scenes paired to GMW region–year labels. For each region and year we select a single acquisition from all candidates

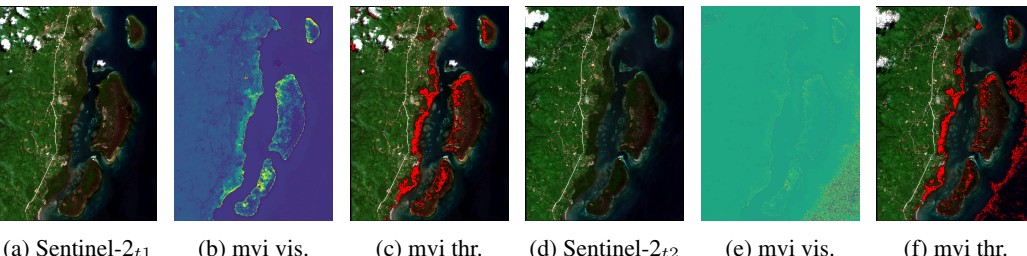

(a) Sentinel-2$_{t1}$  (b) mvi vis.  (c) mvi thr.  (d) Sentinel-2$_{t2}$  (e) mvi vis.  (f) mvi thr.

Figure 1: **mvi visualizations across dates at the same site.** (a) Sentinel-2 RGB at date $t_1$. (b) mvi visualized after 99th-percentile clipping. (c) Binary mask from a fixed mvi threshold 4.5–13.0. (d) Sentinel-2 RGB at date $t_2$ for the same site. (e) mvi visualized with the same preprocessing as in (b). (f) Binary mask using the same fixed threshold as in (c).

using a quality-aware score computed within the region. The score integrates three signals: intersection over union between a composite-trained predictor and the annual mask, Level-2A Aerosol Optical Thickness (AOT) for atmospheric clarity, and cloud fraction. The resulting pairs are more reliable than datasets selected solely by cloud coverage.

Beyond data selection, we introduce an AlphaEarth-guided spectral projection for remote-sensing segmentation. AlphaEarth is a geospatial foundation model trained on global multi-sensor time series that yields compact per-pixel embeddings capturing spatial and temporal context such as climate, season, morphology, hydrology, and human influence Brown et al. (2025). We treat this embedding as location-aware context and learn a lightweight projection that produces a small set of auxiliary channels. These channels act as false bands and are trained to follow the physics of spectral analysis by enlarging between-class contrast while controlling within-class variation. Appending them to the six optical bands lets the segmenter couple same-date reflectance with regional and seasonal priors. The resulting physics-aware design sharpens coastal boundaries and improves robustness to haze, tide, and background variation, and it integrates as a drop-in module with both convolutional and transformer backbones.

**Contributions**

1. We formalize the temporal pairing gap of region–year labels and introduce a quality-aware score that ranks all single-date candidates within a region by composite-based prediction agreement, AOT, and cloud fraction, enabling principled selection without manual alignment.

2. We provide *MANGO*, a globally distributed set of Sentinel-2–GMW single-date pairs with country-balanced coverage, six-band 10 m imagery, per-scene metadata for auditing, and co-registered annual AlphaEarth embeddings, together with region-disjoint 80/10/10 splits.

3. We introduce an AlphaEarth-guided spectral projection that converts frozen embeddings into a small set of learned false bands carrying regional and seasonal context. Trained with a physics-aware separation objective that amplifies inter-class spectral contrast, this module drops into both CNN and transformer based models without modification.

4. We establish reproducible baselines across CNN and transformer based models and show improvements when conditioning on the projected channels.

## 2 RELATED WORKS

**Mangrove detection with spectral indices and classical machine learning**    Early mangrove inventories relied on hand-crafted spectral indices that contrast vegetation against water, soil, and built-up areas by exploiting band physics Tran et al. (2022). The Normalized Difference Vegetation Index uses near-infrared and red to capture chlorophyll vigor Tran et al. (2022). The Enhanced Vegetation Index incorporates the blue band to damp atmospheric and soil influences Tran et al. (2022). NDMI and NDWI variants compare near-infrared with shortwave-infrared, or green with shortwave-infrared, to measure canopy moisture and open water Tran et al. (2022). For turbid coasts, the Man-

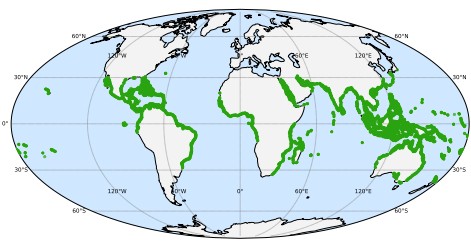 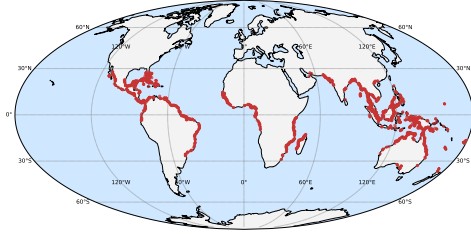

|          |          |
|----------|----------|
| (a) GMW 2020 label footprint. | (b) *MANGO* sampling footprint. |

Figure 2: **Global footprint of labels and samples.** (a) Spatial extent of GMW 2020 label polygons that define region–year supervision. (b) Distribution of *MANGO* sampling regions used to construct Sentinel-2 image–label pairs via the quality-aware selection pipeline.

grove Vegetation Index combines green, near-infrared, and the second shortwave-infrared band to enhance separability of mangrove stands Baloloy et al. (2020). These indices are grounded in the physical response of mangrove canopies, which show strong near-infrared reflectance due to leaf cellular structure, absorption in the red by chlorophyll, and shortwave-infrared sensitivity to water content that helps distinguish canopies from water and mudflats when thresholds are well tuned Tran et al. (2022); Baloloy et al. (2020). At national scale, index features are often paired with classical machine learning on cloud platforms such as Google Earth Engine, for instance a ten-meter Belize inventory that used Sentinel-2 features with random forests to map fine coastal patterns Cissell et al. (2021). Despite these advances, fixed thresholds vary across sensors, seasons, atmospheres, and tidal states, which leads to site-specific retuning and motivates approaches that move beyond static index rules Tran et al. (2022); Baloloy et al. (2020).

**Remote-sensing segmentation models** Deep segmentation has progressed from convolutional encoder–decoders such as U-Net to architectures built on self-attention and modern decoders, and these models transfer effectively to aerial and satellite imagery Ronneberger et al. (2015); Dosovitskiy et al. (2020); Xie et al. (2021); Cheng et al. (2022). In Earth observation, segmentation underpins land cover mapping Wang et al. (2021); Demir et al. (2018), shoreline monitoring Vos et al. (2019), flood delineation Bonafilia et al. (2020), and building extraction Maggiori et al. (2017), which has driven broad adoption across sensors and regions. To mitigate data scarcity and domain shift, recent work explores few-shot segmentation tailored to remote sensing. A representative approach synthesizes novel-class exemplars with an inpainting diffusion pipeline and refines masks to substantially improve low-data performance across backbones Immanuel et al. (2025). In the mangrove domain, studies adapt CNN and Transformer hybrids and report steady gains across diverse coastlines, which underscores the move toward data-driven and generalizable segmentation beyond index thresholds **?**Zhou et al. (2024); Rondon et al. (2023).

## 3 CONSTRUCTING *MANGO*

**Motivation and design principles.** Despite the practical appeal of spectral-index mapping, fixed thresholds lack robustness across sensors, seasons, tidal states, turbidity, and aerosol loads, which forces region-specific retuning and limits scalability. Supervised deep models, in turn, require date-specific image–label pairs, whereas widely used annual products only provide region–year masks without acquisition dates. *MANGO* addresses this temporal pairing gap while maintaining global coverage, geographic balance at the country level, and quality awareness through per-scene metadata that supports auditing and analysis.

### 3.1 DATASET COLLECTION

We tile the GMW extent with a non-overlapping $S \times S$ grid to define regions and, for each region–year, retain all Sentinel–2 L2A acquisitions that sufficiently cover the region while remaining reasonably clear; retained scenes are cropped to the region and assembled as a temporal stack. For a single year, we write the region set as $\mathcal{R} = \{R_k\}_{k=1}^{K}$ and reserve $\mathcal{T}^{R_k}$ for the retained time index set of region $R_k$ ( Equation 1). We denote a Sentinel–2 L2A acquisition at time $t_n$ by $I_{t_n}$, use

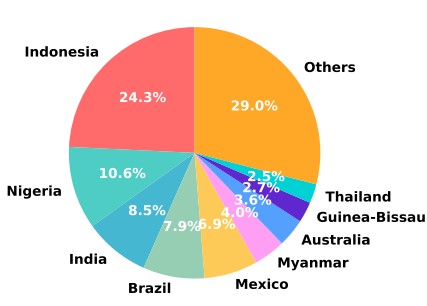 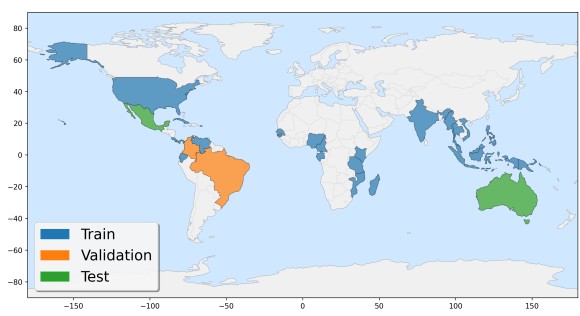

(a) Sampling distribution by country.          (b) Country-disjoint train/validation/test split.

Figure 3: **Sampling distribution and splits.** (a) Proportion of sampled regions across countries in $\mathbf{X}$. (b) Map of the 80/10/10 split at the country level.

$\Omega^{R_k}(I_{t_n}) \in [0, 1]$ for the coverage fraction of $I_{t_n}$ over $R_k$, and use $C(I_{t_n}^{R_k}) \in [0, 1]$ for the cloud coverage over the region in the region-cropped image $I_{t_n}^{R_k}$. We keep all acquisition times that satisfy both coverage and cloud criteria:

$$\mathcal{T}^{R_k} = t_n : \Omega^{R_k}(I_{t_n}) \geq \omega \text{ and } C(I_{t_n}^{R_k}) \leq \kappa, \text{ where } \omega = 0.9, \ \kappa = 0.5. \tag{1}$$

For $t_n \in \mathcal{T}^{R_k}$, we denote the region-cropped image by $I_{t_n}^{R_k}$ and write the per-year image set as

$$\mathbf{X} = \{I_{t_n}^{R_k} \mid R_k \in \mathcal{R}, \ t_n \in \mathcal{T}^{R_k}\} \tag{2}$$

Through this procedure, the sampled regions that constitute $\mathbf{X}$ and their global footprint are summarized in Figure 2. As shown in Figure 3, the dataset exhibits a broad geographic footprint and is partitioned with a country-disjoint split into train, validation, and test; we use an 80/10/10 ratio and hold out validation and test regions that are never seen during training. Each retained acquisition is exported at 10 m and stored as a six–band stack

$$I_{t_n}^{R_k} = [\text{R, G, B, NIR, SWIR1, SWIR2}], \quad I_{t_n}^{R_k} \in \mathbb{R}^{H \times W \times 6}. \tag{3}$$

As shown in Figure 4, the largest between-class gaps in surface reflectance occur in the green, near-infrared, and both shortwave-infrared ranges. We therefore restrict inputs to $[\text{RGB}, \text{NIR}, \text{SWIR1}, \text{SWIR2}]$. This selection follows the same spectral logic used to derive the mvi and related analyses Tran et al. (2022); Baloloy et al. (2020); Cissell et al. (2021).

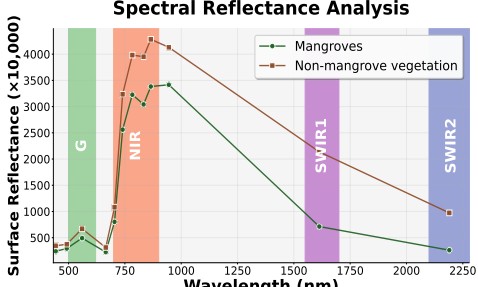

Alongside every $I_{t_n}^{R_k}$ we rasterize the annual GMW polygons on the region grid to obtain the supervision mask $M^{R_k} \in \{0, 1\}^{H \times W}$. We also record per-candidate metadata used during selection: the regional cloud coverage $C(I_{t_n}^{R_k})$ and the aerosol optical thickness $A(I_{t_n}^{R_k})$. In

Figure 4: Surface reflectance analysis for mangroves and non-mangrove vegetation across Sentinel-2 wavelengths.

addition, we provide an annual AlphaEarth embedding for each region, denoted $E^{R_k} \in \mathbb{R}^{H \times W \times 64}$, defined on the same spatial grid as $I_{t_n}^{R_k}$ but without a time index; $E^{R_k}$ summarizes location-dependent priors for the year. The complete field list and formats are provided in Appendix **??**.

## 3.2 DATASET SELECTION

Figure 5 outlines the selection pipeline. The core challenge is to rank same-region candidates without date-specific labels. Direct evaluation against the annual GMW mask is ill-posed, so we adopt a composite-first proxy: for each region $R_k$ we form a temporal median composite $\widetilde{I}^{R_k}$ over $\mathcal{T}^{R_k}$,

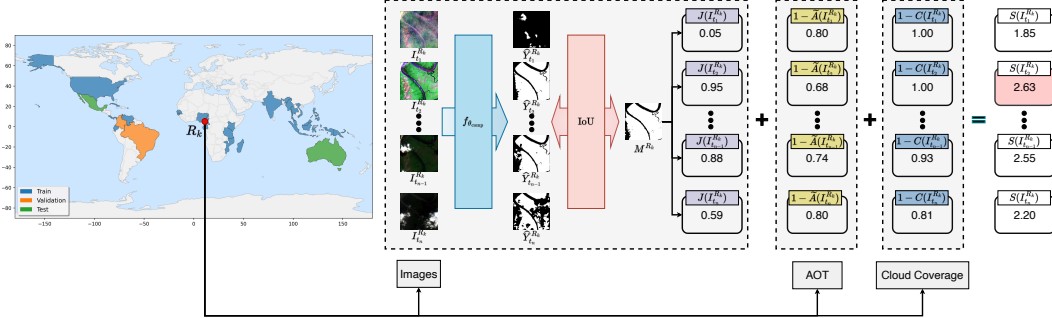

Figure 5: **Dataset selection pipeline.** For each region $R_k$, candidate acquisitions $I_{t_n}^{R_k}$ are assessed with a quality-aware score that integrates predicted alignment to the annual mask, aerosol load, and cloud coverage; the top-scoring acquisition is selected as the training image.

train a lightweight predictor $f_{\theta_{\mathrm{comp}}}$ with learned parameters $\theta_{\mathrm{comp}}$ on $(\widetilde{I}^{R_k}, M^{R_k})$, and run it on each candidate $I_{t_n}^{R_k}$ to obtain probabilities $P_{t_n}^{R_k}$ and a binarized mask $\widehat{Y}_{t_n}^{R_k}$ at a fixed threshold $\tau = 0.5$.

**Prediction term.** We quantify prediction–label consistency for a candidate via the intersection-over-union with the annual mask:

$$J(I_{t_n}^{R_k}) = \mathrm{IoU}(\widehat{Y}_{t_n}^{R_k}, M^{R_k}). \tag{4}$$

**Haze and Cloud term.** We extract AOT $A(I_{t_n}^{R_k})$ and cloud coverage $C(I_{t_n}^{R_k}) \in [0, 1]$ over the region for each candidate. Lower haze and lower cloud should be preferred in the final ranking, which we implement in the score through the factors $1 - \widetilde{A}$ and $1 - C$ after normalizing AOT as

$$\widetilde{A}(I_{t_n}^{R_k}) = \min\left\{\frac{A(I_{t_n}^{R_k})}{1000}, 1\right\}. \tag{5}$$

**Quality-aware score.** We set non-negative weights $w_J, w_A, w_C$ uniformly or tune them on a held-out subset, then define the quality-aware score and the selection rule as follows:

$$S(I_{t_n}^{R_k}) = w_J J(I_{t_n}^{R_k}) + w_A(1 - \widetilde{A}(I_{t_n}^{R_k})) + w_C(1 - C(I_{t_n}^{R_k})), \tag{6}$$

$$t_{\mathrm{best}} = \arg \max_{t_n \in \mathcal{T}^{R_k}} S(I_{t_n}^{R_k}). \tag{7}$$

## 4 ALPHAEARTH-GUIDED SPECTRAL PROJECTION FOR MANGROVE SEGMENTATION

Classical spectral analysis separates targets by forming indices from differences in surface reflectance across bands. For mangrove detection the strongest discriminative gaps arise in near-infrared, shortwave-infrared, and green, and prior work exploits these regimes Tran et al. (2022); Baloloy et al. (2020); Cissell et al. (2021). We translate this physics into learning by drawing location-dependent cues from the AlphaEarth embedding through a lightweight projection layer and appending the resulting channels to the six optical bands. Training encourages the projected channels to enlarge the margin between mangrove and non-mangrove, and the segmentation loss anchors pixel-wise predictions to the ground truth. The design plugs into convolutional and transformer backbones without backbone-specific changes and yields consistent performance gains.

### 4.1 OVERALL PIPELINE

Figure 6 summarizes our method. It integrates in a backbone-free manner with convolutional and transformer segmentation models. Each model receives an input built by concatenating the six

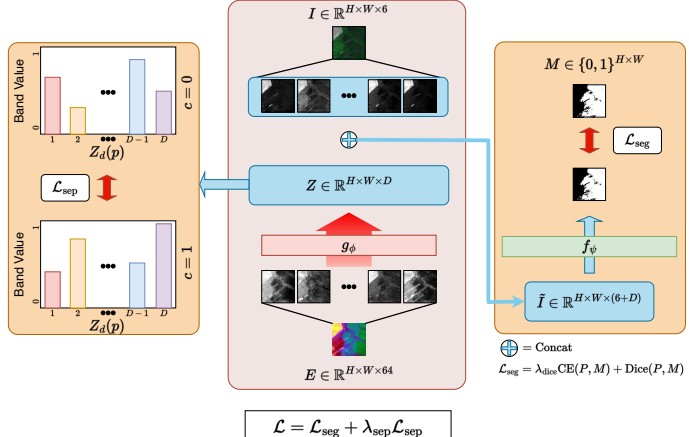

Figure 6: **Overview of the proposed pipeline.** The AlphaEarth embedding is projected into a small set of context channels. These channels are concatenated with the six optical bands, and the resulting stack is fed to the segmentation model, which is trained with a standard segmentation loss. In parallel, a separation loss acts on the projected channels: for c=0 (non-mangrove) and c=1 (mangrove), it increases the difference between their band statistics while keeping each class compact. Training minimizes the sum of the segmentation loss and the separation loss.

optical bands with channels obtained by projecting the AlphaEarth embedding. We optimize the network with a standard segmentation loss and add a separability loss that drives the projected channels to enlarge the margin between mangrove and non-mangrove. The two losses act jointly so that the model exploits same-date spectra while leveraging slowly varying regional priors.

### 4.2 PROJECTION LAYER

In this section we write the selected image $I^{R_k}_{t_{\text{best}}}$ as $I$, the annual AlphaEarth embedding $E^{R_k}$ as $E$, and the mask $M^{R_k}$ as $M$. AlphaEarth supplies $E$ on the same spatial grid as $I$ and, at each pixel, provides a 64-dimensional vector that summarizes location-dependent priors. The projection layer converts this per-pixel vector into $D$ arbitrary channels via a shared linear map and bias, producing task-aligned false bands. Formally,

$$Z = g_\phi(E) = EU + b, \qquad E \in \mathbb{R}^{H \times W \times 64},\ U \in \mathbb{R}^{64 \times D},\ b \in \mathbb{R}^D,\ Z \in \mathbb{R}^{H \times W \times D}. \quad (8)$$

We then concatenate $Z$ with the six optical bands to form the model input, which exposes both same-date spectra and the learned regional context to the backbone:

$$\tilde{I} = \text{concat}(I, Z) \in \mathbb{R}^{H \times W \times (6+D)}. \quad (9)$$

Any segmentation model, CNN-based or transformer-based, consumes $\tilde{I}$ and produces per-pixel logits and probabilities optimized by the losses defined below:

$$\hat{P} = f_\psi(\tilde{I}), \qquad P = \text{Softmax}(\hat{P}). \quad (10)$$

### 4.3 LOSS FUNCTION

The projection layer injects geospatial and seasonal context from AlphaEarth. To make the projected channels $Z$ act as learned false bands that carry this context in a discriminative way, we introduce a separation loss in the spirit of spectral analysis. The objective is to enlarge the gap between the class-conditional distributions while keeping within-class variance small under supervision from $M$. Let $P$ denote the predicted per-pixel class probabilities on $\tilde{I}$. For $c \in \{0, 1\}$ define $\Omega_c = \{p \in \Omega \mid M(p) = c\}$ and per-dimension statistics

$$\mu_{c,d} = \frac{1}{|\Omega_c|} \sum_{p \in \Omega_c} Z_d(p), \qquad \sigma^2_{c,d} = \frac{1}{|\Omega_c|} \sum_{p \in \Omega_c} (Z_d(p) - \mu_{c,d})^2. \quad (11)$$

Table 1: Baselines and AlphaEarth-projected variants on the country-disjoint test set (higher is better). "+A.E" denotes AlphaEarth-guided spectral projection. Formatting: **bold** marks the best score in each column and underline marks the second best. Abbreviations: CNN = convolutional backbone, Trans = transformer backbone, Pre/Rec/Acc = precision/recall/accuracy.

| Grouping | | Method | | IoU(%) | F1(%) | Pre(%) | Rec(%) | Acc(%) |
|---|---|---|---|---|---|---|---|---|
| Setting | Backbone | model | encoder | | | | | |
| SM | CNN | UNet++ | resnet34 | 90.84 | 94.60 | 94.36 | 96.00 | 93.93 |
| | | MAnet | resnet34 | 89.82 | 93.81 | 93.40 | 95.83 | 93.09 |
| | | PAN | resnet34 | 88.98 | 93.34 | 92.59 | 95.80 | 92.55 |
| | Trans | Segformer | mit-b2 | 90.25 | 94.20 | 95.12 | 94.61 | 93.60 |
| | | FPN | pvtv2-b2 | 90.33 | 94.25 | 94.44 | 95.37 | 93.57 |
| | | DPT | vit-b16 | 85.05 | 90.85 | 87.01 | 97.40 | 90.21 |
| | | UPerNet | swin-t | 86.00 | 91.49 | 87.58 | **97.82** | 91.15 |
| Ours | CNN | UNet++ + A.E | resnet34 | 90.62 | 94.37 | 95.58 | 94.49 | 93.91 |
| | | MAnet + A.E | resnet34 | 90.72 | 94.44 | 95.52 | 94.68 | 93.84 |
| | | PAN + A.E | resnet34 | 90.86 | 94.56 | 95.17 | 95.13 | 94.05 |
| | Trans | Segformer + A.E | mit-b2 | **91.27** | **94.88** | 95.54 | 95.30 | **94.28** |
| | | FPN + A.E | pvtv2-b2 | 91.12 | 94.79 | **96.00** | 94.67 | 94.26 |
| | | DPT + A.E | vit-b16 | 86.30 | 91.68 | 88.24 | 97.47 | 91.17 |
| | | UPerNet + A.E | swin-t | 85.94 | 91.22 | 89.51 | 95.38 | 91.36 |

Channel-wise separability is $s_d = \frac{(\mu_{1,d} - \mu_{0,d})^2}{\sigma_{1,d}^2 + \sigma_{0,d}^2 + \varepsilon}$ and we aggregate $l_{\text{sep}} = \frac{1}{D} \sum_{d=1}^{D} s_d$. For stability we bound the separation loss term $\mathcal{L}_{\text{sep}}$ to the range $[0, 1]$ (See Equation 12). Alongside this separation objective we use a standard segmentation loss that combines cross-entropy and Dice loss XXX. The final training objective is expressed as follows:

$$\mathcal{L} = \text{CE}(P, M) + \lambda_{\text{dice}} \text{Dice}(P, M) + \lambda_{\text{sep}} \mathcal{L}_{\text{sep}}, \quad \mathcal{L}_{\text{sep}} = \frac{1}{1 + l_{\text{sep}}/\tau}. \quad (12)$$

The projection layer supplies learned false bands that encode regional and seasonal priors, the segmentation loss supervises the mask, and the bounded separation term shapes the projected channels toward maximal class separation. Hyperparameters and optimization schedules are reported in Appendix B.

## 5 EXPERIMENTS

In this section we compare AlphaEarth-guided models with strong baselines on *MANGO*. The split is 8:1:1 for train, validation, and test at the region level, and the validation and test sets contain regions never seen during training (Fig. 3b). All runs were executed on Ubuntu 20.04 with PyTorch 2.5.0, CUDA 11.5, RTX 4090 24 GB, and an AMD Ryzen Threadripper PRO 5955WX CPU.

### 5.1 BASELINES

We benchmark widely used semantic segmentation architectures from the PyTorch ecosystem spanning CNN and Transformer families. CNN baselines: UNet++ Zhou et al. (2018), MAnet Fan et al. (2020), and PAN Li et al. (2018) with ResNet-34 encoders. Transformer baselines: DeepLabV3+ Chen et al. (2018), SegFormer Xie et al. (2021), FPN Lin et al. (2017), DPT Ranftl et al. (2021), and UPerNet Xiao et al. (2018) using SwinV2, MiT-B2, PVTv2-B2, ViT-B/16, and Swin-T encoders.

### 5.2 QUANTITATIVE RESULTS

Table 1 summarizes country-disjoint test performance. Adding AlphaEarth-guided spectral projection (AE-SP) improves mean IoU from 87.78 to 88.36 and mean F1 from 92.18 to 92.60 across

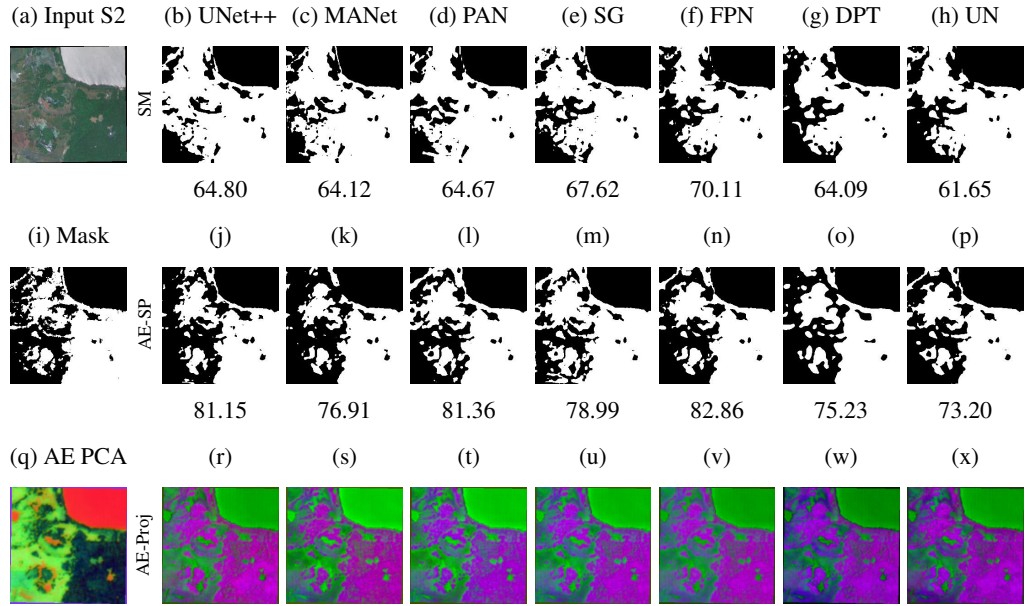

Figure 7: **Qualitative results.** Vertical labels indicate column groups: SM—baseline segmentation outputs; AE-SP—the same backbones with the proposed AlphaEarth-guided spectral projection; and AE_Proj—visualizations of the AlphaEarth embedding projected by AE-SP. Row 1: (a) Input Sentinel-2 RGB; (b–h) predictions from UNet++, MANet, PAN, SegFormer, FPN, DPT, and UPerNet (abbrev.: SG=SegFormer, UN=UPerNet). Row 2: (i) Ground-truth mask; (j–p) AE-SP predictions in the same model order as (b–h). Row 3: (j) AlphaEarth image visualized via PCA (3 components); (r–x) AE-SP–projected AlphaEarth bands, again following the model order in (b–h). IoU is reported beneath each panel.

backbones. Precision increases from 92.46 to 94.02, while recall shows a small average decrease; accuracy rises from 92.00 to 92.56. By family, transformer backbones gain about +0.75 IoU and +0.45 F1 on average, whereas CNNs exhibit smaller but positive deltas. The strongest single systems are the transformer variants with projection, with SegFormer + AE leading IoU/F1 and FPN + AE ranking second. Two models depart from the trend: UNet++ and UPerNet show limited benefit under our setup.

## 5.3 QUALITATIVE RESULTS

Figure 7 presents qualitative results. Row 1 contains baseline predictions; Row 2 shows the same backbones conditioned on the AlphaEarth-guided spectral projection; Row 3 visualizes the AlphaEarth embedding via PCA (left) and the projected AlphaEarth bands (right). Across scenes, conditioning on the projection yields delineations that more closely match the ground-truth mask: coastal boundaries are sharper, small mangrove patches are retained, and commission in turbid water is reduced. The per-panel IoU printed beneath each image tracks these trends.

**Training protocol.** All baselines and guided variants use the same data pipeline, optimization budget, and reporting protocol. Inputs are the six optical bands or the six optical bands concatenated with the $D$ projected channels. Models are trained with the segmentation loss and the bounded separation loss defined in Section 4. Hyperparameters, schedules, and evaluation details are provided in Appendix B.

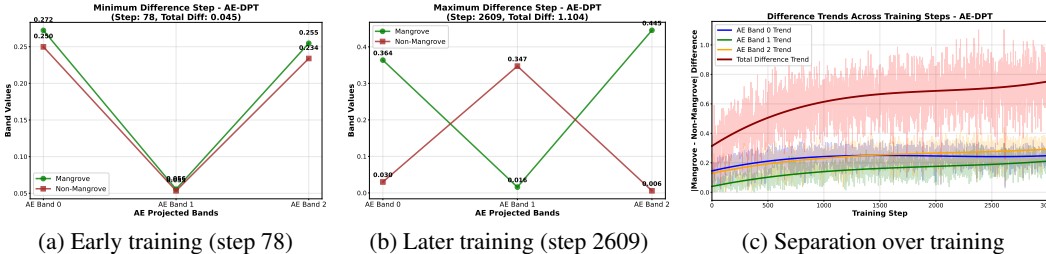

(a) Early training (step 78)    (b) Later training (step 2609)    (c) Separation over training

Figure 8: **Class–separation dynamics of the projected AlphaEarth bands.** Example with a DPT backbone. The AlphaEarth–guided spectral projection is trained with a separation loss that enlarges between–class contrast while controlling within–class variance. (a) At an early step the mangrove and non–mangrove class means across the three projected bands are nearly coincident. (b) As training proceeds the class means diverge and occupy complementary directions across bands, widening the margin. (c) The per–band and total mangrove–non–mangrove differences increase throughout training (solid curves denote smoothed trends), confirming that the separation objective actively shapes the projected channels.

## 6    DISCUSSION

**Dynamics of the projected bands.**    The AlphaEarth-guided spectral projection converts the embedding into a few task-aligned bands and trains them with a separation objective that enlarges the margin between mangrove and non-mangrove while stabilizing within-class variability. This mechanism explains the improvements observed above: the backbone no longer relies solely on single-date reflectance, but fuses it with location-aware priors carried by the projected bands, which reduces ambiguity at tidal shorelines and over turbid water.

Figure 8 examines how this separation emerges during training. Early in optimization (Fig. 8a) the class means across the projected bands are nearly coincident, indicating little discriminative structure. As training proceeds (Fig. 8b) the means diverge and occupy complementary directions across bands, widening the available decision margin. The trajectory plot (Fig. 8c) shows steadily increasing per-band and total differences, confirming that the separation objective actively shapes the projected channels rather than being a by-product of optimization. Consistently with Row 3 of Figure 7, the projected AlphaEarth images display a clear mangrove/non-mangrove contrast—much stronger than the PCA view of the embedding—corroborating that the learned projection is the driver of the qualitative gains.

**Practical advantages.**    The projection is linear and low-dimensional, so it adds minimal computational and memory overhead. Because it augments the input and leaves the backbone unchanged, it transfers across convolutional and transformer families without special engineering. The performance gains hold under a country-disjoint split, which indicates improved behavior when transferring to unseen regions. The data selection pipeline further improves reproducibility by fixing how single-date images are paired with annual labels and by exposing the metadata used to make those choices.

## 7    CONCLUSION

We presented a scalable path to supervised mangrove segmentation that pairs region–year labels with single-date Sentinel-2 scenes and that injects foundation-model priors in a physics-aware manner. The *MANGO* dataset is built by ranking all candidate acquisitions per region with a quality-aware score that integrates prediction agreement, aerosol load, and cloud coverage. On top of these curated pairs, AlphaEarth-guided spectral projection converts geospatial embeddings into learned false bands and trains them to enlarge between-class contrast, delivering consistent gains across convolutional and transformer backbones with minimal overhead.

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

## A  LIMITATIONS AND FUTURE WORK

All supervision ultimately comes from the Global Mangrove Watch annual masks. These masks are produced from satellite data by combining change detection from synthetic aperture radar with optical classification. They are not a census derived from systematic field delineation. Even after selecting the best single-date image per region–year with a quality-aware score that prefers high agreement with the annual mask, low aerosol load, and low cloud cover, residual errors remain. Tidal phase, sensor artifacts, and inherited misclassifications can appear in both the labels and the selected images. Such imperfections may bias both training and evaluation and are a plausible explanation for the modest decrease in recall that accompanies the precision gains. The selection score reduces temporal mismatch and atmospheric effects but cannot certify ground truth. Future releases should include targeted audits with very high-resolution imagery or limited field checks.

Future work will extend selection to multi-date inputs, fuse optical and SAR to reduce cloud sensitivity, and explore joint training with next-generation geospatial foundation models. We hope the dataset, the selection protocol, and the projection module serve as strong baselines for robust, reproducible monitoring of coastal ecosystems at global scale.

## B  TRAINING SETUP

We instantiate backbones from both convolutional and transformer families as reported in the experiments section 5. The projection uses a $1 \times 1$ convolution with learnable bias. We set $D$ to 3 and $\lambda_{\mathrm{dice}}$ to 0.5. We also set $W$ and $H$ to 256. The scale $\tau$ is initialized to 2.0 and updated with an exponential moving average of the running separability during early iterations to maintain stability. The weight $\lambda_{\mathrm{sep}}$ ramps linearly from 0.1 to 0.5 over the first three thousand steps and is then held fixed. When a batch contains a single class the separation term is skipped by setting $\mathcal{L}_{\mathrm{sep}}^{[0,1]}$ to 0. We optimize with AdamW using weight decay $10^{-4}$. To introduce $Z$ smoothly we optionally warm start by freezing the encoder for one to three epochs while training the projection and decoder and then unfreeze all parameters. Settings are shared across backbones unless otherwise stated.

## C  ABLATION STUDY

To validate the scene selection in Equation 7, we reconstruct datasets under four weight settings—J-dominant, AOT-dominant, cloud-dominant, and uniform ($w_J + w_A + w_C = 1$)—and train the same model on each.

Table 2: Ablation of the quality-aware score weights on downstream segmentation. Each row specifies ($w_J$, $w_A$, $w_C$)—the weights on prediction agreement, AOT clarity, and cloud fraction in Equation 6. Results are reported on the country-disjoint test split. Formatting: **bold** marks the best score in each column and underline marks the second best.

| Model | ($w_J$, $w_A$, $w_C$) | IoU(%) | F1(%) | Pre(%) | Rec(%) | Acc(%) |
|---|---|---|---|---|---|---|
| | (1.0, 0.0, 0.0) | 90.45 | 94.35 | **95.03** | 94.88 | **94.69** |
| | (0.0, 1.0, 0.0) | 90.01 | 94.04 | 93.42 | 96.10 | 93.17 |
| | (0.0, 0.0, 1.0) | 90.04 | 94.06 | 93.50 | 95.96 | 93.22 |
| UNet++ | (0.10, 0.45, 0.45) | 90.71 | 94.52 | 94.78 | 95.38 | 93.84 |
| | (0.45, 0.45, 0.10) | 90.77 | 94.55 | 93.94 | **96.36** | 93.83 |
| | (0.45, 0.10, 0.45) | **90.84** | **94.60** | 94.36 | 96.00 | 93.93 |

Table 3: Performance of the composite-trained predictor $f_{\theta_{\mathrm{comp}}}$ used in the selection pipeline. The model is trained on $\left(\widetilde{I}^{R_k}, M^{R_k}\right)$ and evaluated against the annual mask. Metrics are aggregated over regions in the country-disjoint split.

| Model | IoU(%) | F1(%) | Pre(%) | Rec(%) | Acc(%) |
|---|---|---|---|---|---|
| $f_{\theta_{\mathrm{comp}}}$ (lightweight) | 87.72 | 92.34 | 92.30 | 94.56 | 91.56 |

## D  EXPLANATION OF ALPHAEARTH

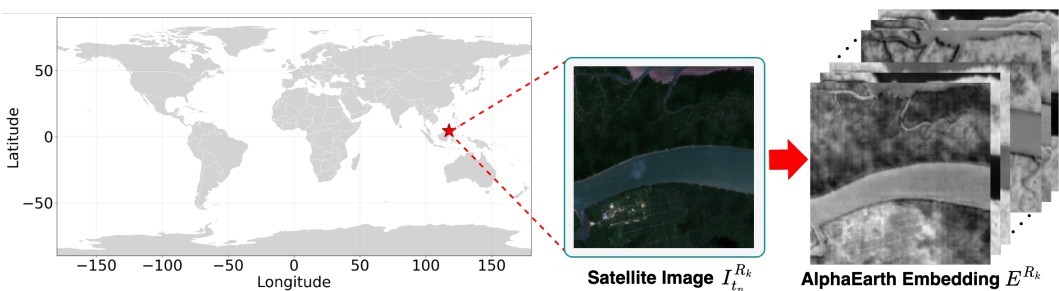

Figure 9: **AlphaEarth embedding overview.** The left panel marks a sampled region on the world map. The center shows the single–date Sentinel-2 scene $I_{t_n}^{R_k}$ selected by the quality-aware procedure. The right shows the co-registered AlphaEarth embedding $E^{R_k}$, provided on the same grid as the optical image.

As shown in Figure 9, for each pixel, AlphaEarth supplies a 64-dimensional vector distilled from global multi-sensor time series that summarizes slowly varying geospatial context, including climate, seasonality, coastal morphology, hydrology, and human activity. In our method the embedding is kept frozen and later projected into a few task-aligned bands that are concatenated with the six optical channels for segmentation.

