# OpenReview forum: "MANGO: MANGROVE GLOBAL OBSERVATIONS –A DATASET AND BENCHMARK"
_ICLR.cc/2026/Conference — ICLR 2026 Conference Withdrawn Submission_

### Official Review · Reviewer_uCpn · 2025-10-22

**Soundness:** 3
**Presentation:** 3
**Contribution:** 3
**Rating:** 6
**Confidence:** 4

**Summary:**

This paper tackles the critical challenge of monitoring mangroves, where long-term annual survey maps often don't align well with specific, single-day satellite images. To address this, the authors first introduce MANGO, a new global dataset, for which they developed a principled selection pipeline that automatically picks the highest-quality and most representative single-date satellite image to pair with its corresponding annual label. Secondly, they propose a novel method that enhances segmentation models by injecting rich, location-aware context from a geospatial foundation model called AlphaEarth. This context is intelligently projected into a few new "false bands" that are added to the original image, providing the model with crucial background knowledge about climate and geography and leading to more accurate results with minimal claimed computational overhead.

**Strengths:**

Originality: The paper demonstrates originality on two distinct fronts. Firstly, it moves beyond simply presenting another dataset and instead offers a novel and principled solution to the pervasive "temporal pairing gap" in remote sensing. The formulation of the quality-aware score which combines prediction agreement from a composite-trained model with atmospheric and cloud conditions is a creative and robust methodology. Secondly, the proposed AlphaEarth-guided spectral projection is an original approach for model enhancement. Rather than performing complex model fusion, it injects rich geospatial priors from a foundation model as learned "false bands," guided by a physics-inspired separation loss.
Quality: The quality of this work is good, primarily demonstrated through its rigorous and thorough empirical validation. The experimental design is sound, most notably in its use of a country-disjoint split for the dataset. This is a gold standard for evaluating geospatial models, as it provides a much more realistic assessment of a model's ability to generalize to entirely new geographic regions, thereby significantly strengthening the credibility of the reported results. Furthermore, the authors benchmark their method against a wide array of strong and diverse baselines from both the CNN and Transformer families, proving the broad applicability of their approach.
Clarity: The paper is clear and well-written. The authors do a job of articulating a complex, domain-specific problem in a way that is accessible to a broader machine learning audience. The motivation is established from the very beginning with a simple yet powerful visual example in Figure 1. The entire methodology, from the data selection pipeline (Figure 5) to the model architecture (Figure 6), is supported by clean and informative diagrams that greatly aid in comprehension. The logical flow of the paper is seamless, guiding the reader from the problem definition through the proposed solutions and to the validating experiments.
Significance: The significance of this paper is substantial and multifaceted. For the specific domain of environmental science and remote sensing, the release of the MANGO dataset provides a much-needed, high-quality benchmark for the critical task of monitoring mangrove ecosystems, which are vital for carbon storage and coastal protection.

**Weaknesses:**

1. The paper asserts that the proposed projection module adds "minimal computational overhead," which is presented as a key advantage. However, this claim is not substantiated with any quantitative evidence. The term "minimal" is subjective and lacks the scientific rigor expected. I think the authors should add a table or a paragraph in the appendix quantifying this overhead. Specifically, they should report the percentage increase such as total model parameters, floating-point operations (FLOPs), and average inference time per image for a representative model (e.g., SegFormer) before and after adding the AlphaEarth projection module. This would provide a clear and objective measure of the method's efficiency.
2. The reported improvements in Table 1 are modest, and a closer inspection reveals a potentially concerning trade-off. I noticed that the recall metric consistently decreases for a majority of the tested backbones after the introduction of the A.E. module, including all three CNN models and two of the four Transformer models. Given that the gains in other metrics like IoU are also marginal, this negative impact on recall makes the overall benefit less clear-cut and raises questions about the method's practical necessity. Furthermore, the table appears to show single-run results, which is insufficient for drawing robust conclusions about these subtle changes. I strongly suggest that the authors report the mean and standard deviation across 3-5 runs with different random seeds. If the improvements are not statistically significant after multiple runs, and considering the negative impact on recall, the authors should honestly discuss whether the added complexity of the AlphaEarth projection module is truly justified.
3. The Related Work section misses several important points. First, the paper does not discuss other mangrove datasets. The authors should at least list MANGO alongside other datasets in a table and provide a comparison of key attributes. Besides, while mentioning the concept of geospatial foundation models, there is no detailed comparison explaining why AlphaEarth was chosen over openly available alternatives like Prithvi (from NASA-IBM), SatCLIP, Clay, or Presto, especially when some of these models have similar capabilities and are more easily reproducible.
4. The ablation study in Table 2, which investigates the weights of the quality-aware score, is provided only for the UNet++ backbone. To demonstrate that these findings are consistent and not specific to a single architecture, a more comprehensive ablation should test this component across multiple backbones. Furthermore, I am curious about the choice to set the projection dimension D=3 (Appendix B). The paper presents this as a fixed value, and a brief justification for this hyperparameter selection would be beneficial to clarify this important design choice.
5. Some related references are recommended to your related works, such as 10.1109/MGRS.2024.3479871, https://doi.org/10.1016/j.isprsjprs.2021.01.008, and https://arxiv.org/abs/2507.14697)

**Questions:**

1. On “minimal computational overhead”: Please quantify this claim. For a representative backbone (e.g., SegFormer), report the percentage increase in parameters, FLOPs, and average per-image inference time before vs. after adding the AlphaEarth projection module (and, if possible, memory usage).
2. On statistical robustness and recall trade-off: Several models show a decrease in recall while overall gains are modest. Please report mean ± std over 3–5 runs with different seeds. If improvements are not statistically significant and recall drops persist, discuss whether the added complexity is justified.
3. On Related Work coverage: Please add a comparison table of mangrove datasets and position MANGO among them (coverage, resolution, etc.). Also, explain why AlphaEarth was chosen over openly available alternatives (e.g., Prithvi, SatCLIP, Clay, Presto), especially regarding capabilities and reproducibility.
4. On ablations and hyperparameters: The weight ablation of the selection score is shown only for UNet++. Please extend to multiple backbones to verify consistency. Also, briefly justify the choice of projection dimension D=3.
5. Please clarify the exact data split: is it country-disjoint, region-disjoint, or both? Figure 3(b) says “country-disjoint”, while the paper also mentions “region-disjoint 80/10/10” and “8:1:1 at the region level”. Please specify which protocol is used for train/val/test and which results correspond to which split.

---

### Official Review · Reviewer_U2Ve · 2025-10-24

**Soundness:** 2
**Presentation:** 3
**Contribution:** 1
**Rating:** 2
**Confidence:** 4

**Summary:**

The paper introduces a new dataset and method. The dataset is a collection of Sentinel-2 images of globally sampled mangroves (coastal forests). The dataset, called Mango, is important because mangroves store carbon and thus should be monitored — and no existing mangrove dataset is sufficient. The method fetches AlphaEarth embeddings (64-d pixel-wise embeddings of the Earth from Google DeepMind) and linearly projects down to C channels. These C channels are stacked on the image and fed to a segmentation model for prediction. The method minimizes loss between segmentation labels and predictions, and maximizes the separability between mangrove and non-mangrove classes of the C channels that were linearly projected from AlphaEarth embeddings.

**Strengths:**

- Well written
- A mangrove dataset like this could be useful to the ML community as a benchmark for remote sensing foundation models
- The use of AlphaEarth embeddings as inputs for a segmentation model is a good idea and shows promise

**Weaknesses:**

- As far as I can tell, there is no ablation on the introduced loss term: the separation loss. To me this ablation is important because a simpler strategy would just train the AlphaEarth embedding projector end-to-end (i.e., only use the final segmentation loss). I see additional experiments in the appendix but this ablates the dataset not the loss terms. If I missed this experiment please point me to it.
- Key dataset characteristics are unclear to me. For example, what is the size of the Sentinel-2 images? The appendix mentions W and H are 256, but is this the size of all images provided by the dataset or did the authors crop down to 256 from a larger dimension? Also how many of these H x W x 6 Sentinel-2 images are there in the dataset? I see that the split is 80/10/10, but I can't find the number of samples. My apologies if I missed these details.
- Training details. The authors train all baselines themselves, as this dataset is new this is needed of course. However, I am concerned that a single training recipe is provided in Appendix B. The authors should do a hyperparameter sweep with equal budget for all methods to ensure fairness. Furthermore, the proposed method introduces additional hyperparameters which appear tuned — but how were they tuned?

**Questions:**

Please see weaknesses.

---

### Official Review · Reviewer_2Fbw · 2025-10-24

**Soundness:** 1
**Presentation:** 2
**Contribution:** 2
**Rating:** 4
**Confidence:** 5

**Summary:**

The authors propose MANGO, a spatially global dataset of Sentinel-2 data aligned with Global Mangrove Watch (GMW) annotations for mangrove monitoring, with labels and scene quality assessment through a quality score. The dataset includes geospatial splits to train and evaluate machine learning models.
They introduce a spectral projection method to leverage AlphaEarth embeddings as additional context during training. The training objective includes a segmentation loss and a physics-aware separation objective. The authors conducted experiments showing that the AlphaEarth embeddings, the linear projection, and the combination of losses have a positive impact on the downstream task of mangrove segmentation.

**Strengths:**

1. Mangrove monitoring at global scale is an important research topic to better understand vegetation carbon storage and climate change impacts on coastal regions.

2. The proposed MANGO dataset with aligned Sentinel-2 and GMW annotations is a novelty at this scale. The proposed geospatial splits are relevant and will avoid spatial autocorrelation.

3. The proposed quality score used to build MANGO is relevant for the application and could be used in other remote sensing applications such as forest cover estimation.

4. The authors propose to add additional features from a foundation model (AlphaEarth) to convolutional or transformer backbones to improve the downstream application of mangrove segmentation.

**Weaknesses:**

1. There is a lack of related work on existing mangrove monitoring datasets [1, 2], methods for mangrove segmentation with remote sensing [3, 4, 5, 6, 7], quality assessment indicators for mangrove mapping [8], and their comparison with the proposed dataset and proposed methods.

2. There is a lack of related work on remote sensing foundation models that could potentially provide embeddings useful for the task [9, 10, 11, 12, 13, 14], and their comparison with AlphaEarth. Since AlphaEarth has not been published and its comparison with other competing methods is not exhaustive, it is not yet clear why this model has been chosen.

3. There is no proper analysis of the optimal dimension D of the projection layer and its properties. There is a lack of ablation studies to define D which could be any value between 1 and 64. Even though the authors claimed these embeddings carry regional and seasonal context (L.79, L.94, L.316, L.355), there is no analysis supporting it.

4. The separation loss is poorly explained in Section 4.3 and Figure 6. One may suggest that including Figure 8 and related discussion earlier in the text to better understand its formulation. That being said, the overall impact of the separation loss is not clear in the experiments: there is no proper ablation study conducted with and without this loss term.

5. There is a lack of experiments and ablation studies to better understand the impact of AlphaEarth features. 1/ One may wonder if the Sentinel-2 images are actually useful: What would be the performance of using the AlphaEarth embeddings only? 2/ Even if AlphaEarth has shown good performance compared to a few other methods, what would be the impact of using state-of-the-art remote sensing foundation models pretrained on large-scale remote sensing and meteorological datasets [9, 10, 11, 12, 14], and sometimes used for downstream vegetation monitoring applications [12, 13]?

6. There is no standard deviations provided with the quantitative results (Table 1). One could question the robustness of the results as they are numerically close.


**References:**

[1] A. Twomey & C. Lovelock, Global spatial dataset of mangrove genus distribution in seaward and riverine margins. In Scientific Data 2024.

[2] M. Jia et al., Mapping global distribution of mangrove forests at 10-m resolution. In Science Bulletin 2023.

[3] D. Wang et al., Evaluating the Performance of Sentinel-2, Landsat 8 and Pléiades-1 in Mapping Mangrove Extent and Species. In Remote Sensing 2018.

[4] D. Hicks et al., Mangrove Ecosystem Detection using Mixed-Resolution Imagery with a Hybrid-Convolutional Neural Network. In NeurIPS Workshop 2020.

[5] H. Dong et al., MangroveSeg: Deep-Supervision-Guided Feature Aggregation Network for Mangrove Detection and Segmentation in Satellite Images. In Forest 2024.

[6] M. Guo et al., ME-Net: A Deep Convolutional Neural Network for Extracting Mangrove Using Sentinel-2A Data. In Remote Sensing 2021.

[7] Y. Zhang et al., MW-SAM:Mangrove wetland remote sensing image segmentation network based on segment anything model. In IET Image Processing 2024.

[8] Kathiroli R et al., Spatiotemporal analysis of mangroves using median composites and convolutional neural network. In Scientific Reports 2025.

[9] G. Tseng et al., Lightweight, Pre-trained Transformers for Remote Sensing Timeseries. In ArXiv 2024.

[10] A. Fuller et al., CROMA: Remote Sensing Representations with Contrastive Radar-Optical Masked Autoencoders. In NeurIPS 2023

[11] G. Tseng et al., Galileo: Learning Global & Local Features of Many Remote Sensing Modalities. In ICML 2025.

[12] G. Astruc et al., AnySat: One Earth Observation Model for Many Resolutions, Scales, and Modalities. In CVPR 2025.

[13] N. Bountos et al., FoMo: Multi-Modal, Multi-Scale and Multi-Task Remote Sensing Foundation Models for Forest Monitoring. In AAAI 2025.

[14] K. Wu et al., A semantic-enhanced multi-modal remote sensing foundation model for Earth observation. In Nature machine intelligence 2025.

[15] C. Brown et al., AlphaEarth Foundations: An embedding field model for accurate and efficient global mapping from sparse label data. In ArXiv 2025.

**Questions:**

**Questions:**

1. L. 232: What is the architecture of the lightweight predictor?

2. L. 248: It is not clear how the non negative weights are defined (uniformly or tuned). What are their values?

3. Why using AlphaEarth versus other remote sensing foundation models (see Weaknesses section)?

4. How can we ensure the proposed context (output of the projection layer) actually contains regional and/or seasonal context as claimed by the authors (see Weaknesses section)?

5. What would be the performance of the convolutional and transformer backbones if they were using the raw AlphaEarth embeddings only?


**Comments:**
- L. 161: Notation are heavy while it does not seem necessary. Why using $R_k$ instead of $k$? Also, $t_n$ is confusing since $n$ is not defined, why using $t_n$ instead of $n$ only?

- Equation 1: How $\omega$ and $\kappa$ have been defined?

- L. 210: It seems the appendix is missing.

- Figure 4: The "non-mangrove vegetation" category could include diverse vegetation types that will have different spectra, e.g. tree canopy versus peatland. It would be appreciated to distinguish different spectra examples instead of a single "non-mangrove vegetation".

- Figure 5, 6 and 8: Please increase the size of the text which is too small to be read without a significant zoom.

- Figure 6: Please detail all acronyms and notations in the caption. Typo on $\lambda_\text{dice}$ applied to CE instead of Dice in the segmentation loss. What is the value of D?

- L. 350: Typo XXX.

- It would be appreciated to add the limitations and future works in the main text since it's a relevant and insightful section.

---

### Note · Authors · 2025-11-20

**Comment:**

We sincerely appreciate the time and effort you dedicated to reviewing our manuscript. Your thoughtful feedback has provided us with clear direction on how to improve the clarity and impact of our work.

To fully address your suggestions and conduct the necessary additional experiments, we have decided to withdraw the paper at this stage. Thank you again for your valuable insights and for helping us strengthen our research.

**Withdrawal Confirmation:**

I have read and agree with the venue's withdrawal policy on behalf of myself and my co-authors.